# CookingCLIP: Learning a Contextualized Multimodal Embedding from Instructional Cooking Videos for Zero-shot Recipe Generation

## Abstract

Cooking is one of the oldest and the most common human activities in everyone's daily life. Instructional cooking videos have also become one of the most common data sources for multi-modal visual understanding researches, because compared to other general domains, cooking videos: 1. not only have a significantly stronger cross-modal dependency between the speech texts and their corresponding visual frames at each individual step, 2. but also have a significantly stronger cross-context dependency between sequential steps along their temporal dimensions, making it an ideal domain for contextualized multi-modal embedding. This paper proposes CookingCLIP, which introduces the latest CLIP (Contrastive Language-Image Pre-training) embedding from the general domain into the specific domain of cooking understanding, and makes two adaption upon the original CLIP embedding for better customization to the cooking understanding problems: 1. from the upstream perspective, we extend the static multi-modal CLIP embedding with a temporal dimension, to facilitate context-aware semantic understanding; 2. from the downstream perspective, we introduce the concept of zero-shot embedding to sequence-to-sequence dense prediction domains, facilitating CLIP being not only capable of telling "Which" (cross-modal recognition), but also capable of telling "When" (cross-context localization). Experiments conducted on two challenging cooking caption generation benchmarks, YouCook and CrossTask, demonstrate the effectiveness of the proposed embedding. The code will be released.

## 1 Introduction

The close relationship between cooking and human life has not only led to a large amount of data sets widely available for many NLP / CV / mulit-modal researches, but has also posed many real-world challenges for theoretical deep learning researchers on how to solve problems that are more closely related to the real need of human life. Up to 2023, cooking recipes/videos have accounted for one of the largest proportion of data sets in many NLP/CV sub-areas, ranging from text/vision-only name entity recognition Zhang et al. (2022), knowledge extraction Wu et al. (2022); Papadopoulos et al. (2022); Xu et al. (2020), planning Lu et al. (2023), summarization Koupaee & Wang (2018); Narasimhan et al. (2022), generation Udhayanan et al. (2023); Noever & Noever (2023), to multi-modal retrieval Alikhani et al. (2022); Voutharoja et al. (2023); Tian et al. (2022), grounding Fang et al. (2023); Tan et al. (2023); Bao et al. (2021), question answering Yang et al. (2022), and many well-known benchmarks focused on cooking-related topics have been proposed in the last few years, such as Recipe1M Salvador et al. (2017), RecipeQA Yagcioglu et al. (2018), YouCook2 Zhou et al. (2018), CrossTask Zhukov et al. (2019a), COIN Tang et al. (2019), Breakfast Kuehne et al. (2014), TACOS Regneri et al. (2013), to name a few.

### 1.1 Motivation from a domain-specific perspective

However the abundance and richness in terms of both quantity and diversity of the cooking data proposed, so far the majority of which have been serving only as a composing part to train and evaluate the large-scale multi-modal models in the general domain, whose adaption to specific cooking problems have been left very much under-explored: the question about how much to their

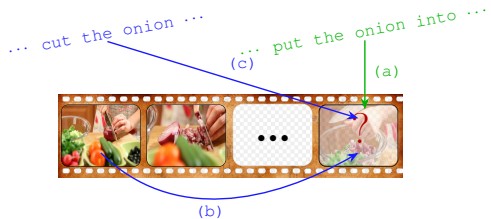

There are three sources of self-supervision co-existing in many instructional video + language cooking data sets:

(a). the static cross-modal self-supervision from texts to frames at synchronous temporal locations: " ... put the onion into ... " → the vision of the onion after being cut;

(b). the dynamic cross-context self-supervision from earlier frames to frames at asynchronous temporal locations: the vision of the original onion before being cut → the vision of the onion after being cut;

(c). the joint cross-context + cross-modal self-supervision from earlier texts to frames at asynchronous temporal locations: " ... cut the onion ... " → the vision of the onion after being cut;

where (b) and (c) remain as a major gap between existing static multi-modal embedding and human-level visual understanding.

Figure 1: The motivation behind our proposed contextualized multi-modal embedding.

full potential can the most advanced multi-modal learning techniques in general domains, such as the multi-modal transformers Shvetsova et al. (2022); Sun et al. (2019), the multi-modal contrastive pre-training Chen et al. (2023), the multi-modal in-context prompting Wang et al. (2022), and even the multi-modal large language models Lu et al. (2023), benefit the cooking related problems has not been raised, explored, nor comprehensively answered yet.

## 1.2 MOTIVATION FROM AN OPEN-DOMAIN PERSPECTIVE

Moreover, cooking also serves an unique and irreplaceable domain where the study of an open-domain contextualized multi-modal embedding is the most suitable to start out, because of its obviously stronger cross-modal + cross-context joint complementary cues than other domains (Figure 1). Although the multi-modal embedding and the text-only contextualized language modeling have each achieved historical progress in their respective fields, research on cross-context + cross-modal joint embedding in the open domain is still very challenging Driess et al. (2023); Li et al. (2023a); Alayrac et al. (2022); Wang et al. (2023) due to too much noise in the raw data Gao et al. (2022) and the breakdowns caused by catastrophic forgetting in large-scale training Li et al. (2022); Srinivasan et al. (2022).

Therefore, starting from a low-noise narrow domain provides a promising path towards a successful contextualized multi-modal embedding that could have potentially benefit research areas from both perspectives.

## 1.3 CONTRIBUTION FROM A DOMAIN-SPECIFIC PROSPECT

As a small step towards more comprehensive cooking understandings in the era of large language models, we propose a quasi-new benchmark for the purpose of a fair formal evaluation, named as cooking recipe generation (CRG), however born from the traditional dense video caption (DVC) Yang et al. (2023a); Zhu et al. (2022) benchmark, their exists two major differences in-between:

1) The traditional DVC reports n-gram matching scores like BLEU-4, METEOR, CIDEr, and ROUGE, which are not originally designed for evaluating semantic similarities, while our proposed CRG replaces them with more advanced Bert-score Zhang* et al. (2020) & Clip-score Hessel et al. (2021), which are more widely employed in modern text generation community;

2) The traditional DVC solutions are neither zero-shot transferable nor open-vocabulary, and need to be fine-tuned on a closed set of pre-defined vocabularies with manual annotations, while our proposed CRG inherits all major advantages of modern multi-modal pre-trained models, which is zero-shot, open-vocabulary, and fully free from manual supervision.

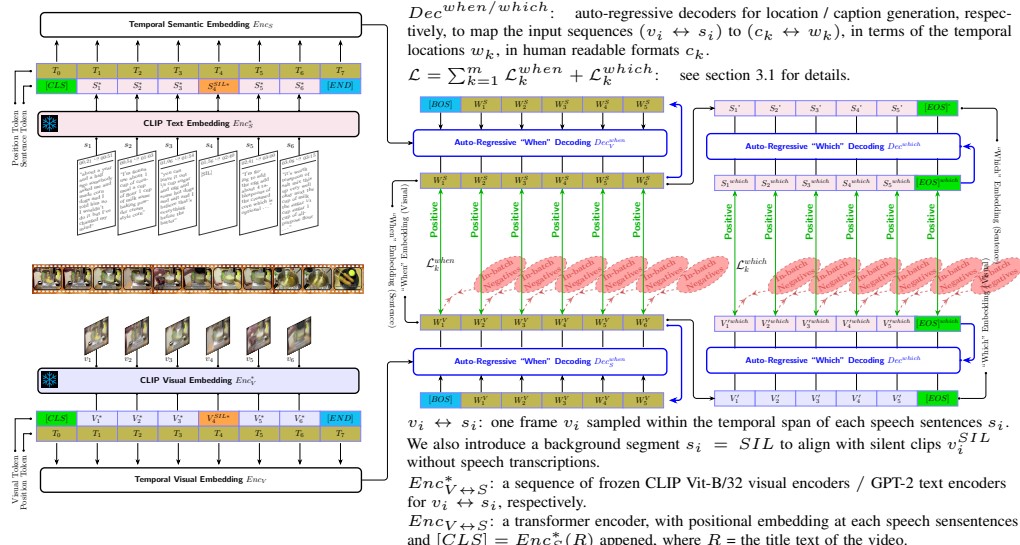

Figure 2: Training illustration of our proposed cross-modal ("Which") + cross-context ("When") multi-modal embedding.

## 1.4 CONTRIBUTION FROM AN OPEN-DOMAIN PERSPECTIVE

As a small step towards a more comprehensive embedding, we propose to formulate the contextualized sequence-to-sequence dense prediction task as a joint semantic localization + embedding problem, where an instructional video $R$ of length $T$ (frames $V_R$ and utterances $S_R$) is mapped to:

- a discrete output sequence of semantic labels $C_R$ (referred to as "Caps" in our context), in a temporal order, and in the forms of natural language descriptions ("Which"),
- along with their temporal localizations $W_R$, in the form of time stamps ("When").

By decoupling the localization and caption decoders, we eliminate the need of the beam search decoding in traditional sequential prediction, and reduce the computing complexity to the same order as a static image caption problem, with only an additional linear coefficient. Experimental results show that our method not only outperforms the current state-of-the-art zero-shot baselines, but also inspiringly catches up with fully supervised methods with only a small gap despite that our results are obtained under a zero-shot setup.

## 2 RELATED WORKS

We gain our inspirations mainly from the well-known CLIP Radford et al. (2021) embedding, who matches fully supervised baselines without the need for any manual supervision. CLIP ensembles human intelligence in almost every aspect, which is zero-shot, open-vocabulary, scalable and generalizable, except one last major difference, lying in that humans do not gain the common sense of this world by static images. Instead of static image - language pairs, humans recognize objects and understand semantics within a dynamic context, along temporal and spatial dimensions.

### 2.1 ZERO-SHOT MULTI-MODAL EMBEDDING

There are efforts which do have noticed the absence of the temporal contextualization in CLIP, and extend the idea of zero-shot embedding to the spatiotemporal space, like VideoCLIP Xu et al. (2021), CLIP-VIP Xue et al. (2023), TempCLR Yang et al. (2023b) and so on, but they are still encoding-only models and a lack of decoders prevents them from adapting to a challenging contextualized cooking understanding problem, where an advanced task usually requires a series of interdependent sub-steps carried on in a progressive step-by-step way.

## 2.2 ZERO-SHOT MULTI-MODAL TEXT GENERATION

A contrary line of works try to cast multi-modal embedding back to their high-level semantics in terms of natural language, in a zero-shot way, by either aligning with Tewel et al. (2022); Mokady et al. (2021); Li et al. (2023b) or prompting Zeng et al. (2023) the existing moderate / large language models. Corresponding to the lack of temporal contextualization, in the previous section 2.1 from an upstream perspective, the main difference between this line and our work is a lack of temporal localization, from a downstream perspective, making them also not well customized to a cooking understanding problem.

## 2.3 TRADITIONAL DENSE VIDEO CAPTION GENERATION

Another related line of works are capable of both upstream contextualization and downstream localization, with the help of seq2seq architectures Yang et al. (2023a); Zhu et al. (2022), however they are not zero-shot scalable, needs to be fine-tuned, and very much time consuming with traditional beam search decoders. To the best of our knowledge, we are not only the first to introduce the concept of zero-shot embedding to a sequence-to-sequence dense prediction task, but also owns an additional advantage not shared by the mentioned methods of Yang et al. (2023a); Zhu et al. (2022), which is highly cost-effective: by decoupling the traditional beam search decoder into two separate decoders for localization and generation, respectively, the exponential computational complexity of traditional sequence-to-sequence prediction problems is reduced to linear.

## 3 ZERO-SHOT RECIPE GENERATION

From the upstream perspective, our model facilitates "Which" + "When" joint embedding:

- not only captures the text $\leftrightarrow$ visual cross-modal relationships ("Which"), similar to the static CLIP embedding,
- but also captures the cross-context temporal relationships ("When") between frame $\leftrightarrow$ sentence sequences,

All necessary pre-training data of $R$, $T$, $V_R$, $S_R$, $T_R$ are fully free from human supervision and widely available from the internet: see Appendix for details on data preparations.

Our architecture is very simple, composed only of a pair of original CLIP encoders for each modality, each of which being extended with a pair of auto-regressive decoders for localization and caption generation, respectively. Our training and inference processes are also very simple, which are illustrated in Figure 2 and 3, respectively.

## 3.1 TRAINING

We initialize our seq2seq models $Enc$ - $Dec^{when}$ / $Dec^{which}$ for both modalities with a traditional unsupervised temporal embedding method Kukleva et al. (2019), and then apply a CLIP-like contrastive loss at each auto-regressive step, to pull positive pairs of contextualized embedding together and to push negative pairs of the other batch instances apart:

$$\mathcal{L}_k^{when} = -\log \frac{\exp\left(\mathsf{sim}\left(\mathsf{w}_k^{\mathsf{V}} \leftrightarrow \mathsf{w}_k^{\mathsf{S}}\right)/\tau\right)}{\sum_{b=1}^{batch=B} \mathbb{1}_{[b \neq k]} \exp\left(\mathsf{sim}\left(\mathsf{w}_k^{\mathsf{V}} \leftrightarrow \mathsf{w}_b^{\mathsf{S}}\right)/\tau\right)}$$

$$\mathcal{L}_k^{which} = -\log \frac{\exp\left(\mathsf{sim}\left(\mathsf{v}_k^{\mathsf{which}} \leftrightarrow \mathsf{s}_k^{\mathsf{which}}\right)/\tau\right)}{\sum_{b=1}^{batch=B} \mathbb{1}_{[b \neq k]} \exp\left(\mathsf{sim}\left(\mathsf{v}_k^{\mathsf{which}} \leftrightarrow \mathsf{s}_b^{\mathsf{which}}\right)/\tau\right)}$$

where $sim(\leftrightarrow)$ is the cosine similarity, and $\tau$ is the temperature hyper-parameter set to $1/100$ in our experiments.

To facilitate customizable temporal granularity of semantic localization, we insert a mandatory $W_m = [EOS]$ when the sum of the absolute values of the first $(m-1)$ locations $\left\|W_k\right\|$ exceeds a predefined threshold $\lambda$:

$$\sum_{k=1}^{m-1} \left\|W_k\right\| > \lambda \tag{1}$$

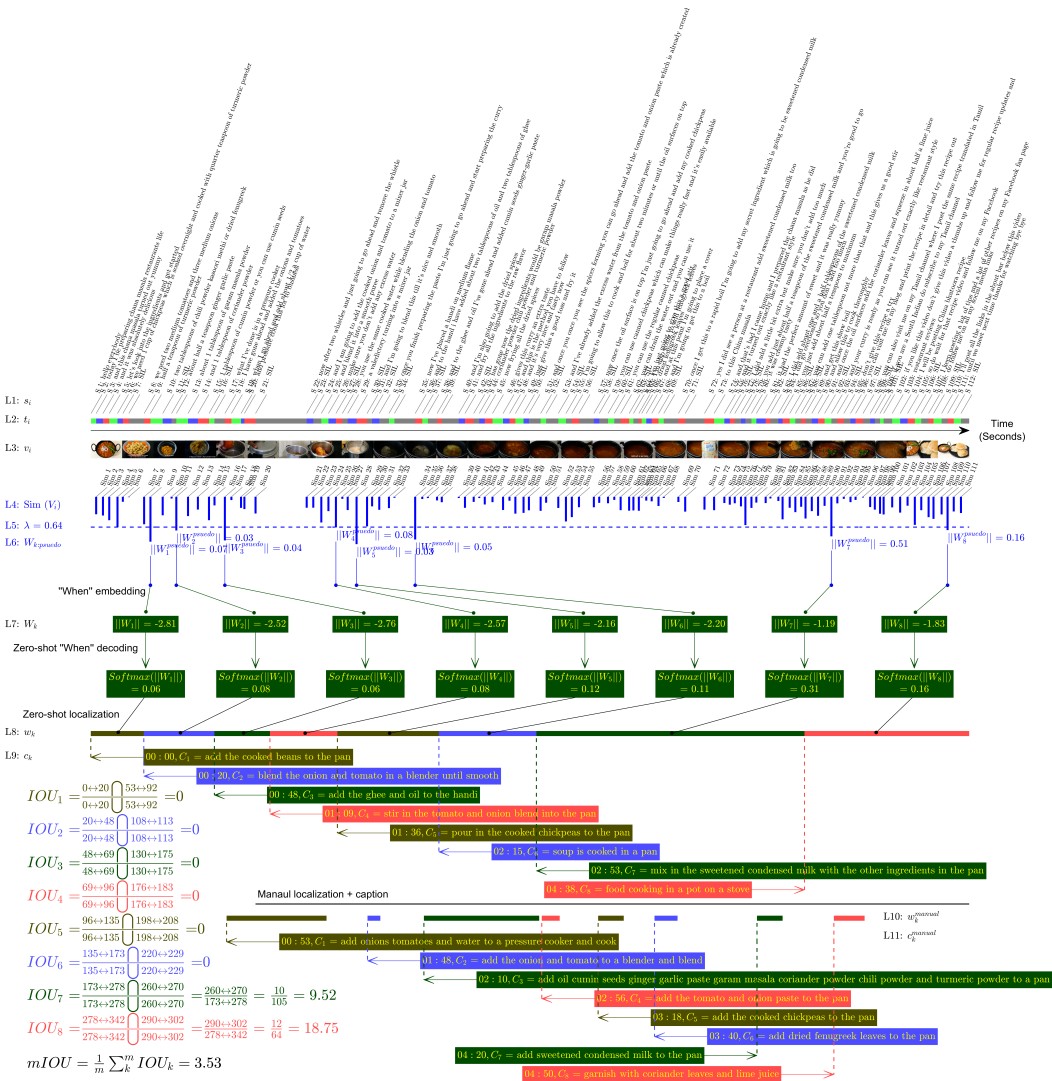

The sample ($\mathtt{vid} = \mathtt{LeCwqp8Bic8}$) lasts 342 seconds with $n = 112$ clip / sentence pairs, including 34 silent clips $v_i^{SIL}/s_i = SIL$ without virtual utterances. It is manually annotated with $m = 8$ captions describing major steps towards the a cooking task titled as $R = $ "$\mathtt{restaurant\ style\ channa\ masala}$". The horizontal colored lines, from top to bottom, are: 1. L1 $\sim$ L3: the input $(s_i \leftarrow t_i \rightarrow v_i)$; 2. L4 $\sim$ L6: the unsupervised hyper-parameter for initialization; 3. L7 $\sim$ L9: the zero-shot temporal semantic localization & caption generation $(w_k \leftrightarrow c_k)$; 4. L10 $\sim$ L11: the manual location & captions $(w_k^{manual} \leftrightarrow c_k^{manual})$, provided by the data set.

Figure 3: Qualitative generation results of a random sample from YouCook2 validation split, compared with manual annotations provided by the data set.

where each $\left\| W_k \right\|$ is forced to be positive definite, and mapped to a closed interval of $(0, 1]$, see Figure 3 for a detailed illustration.

Upon the prediction of $W_m = [EOS]$, $\mathcal{L}_k$ is aggregated across both modality and all autoregressive steps $k$, to facilitate a global minimum:

$$\mathcal{L} = \sum_{k=1}^{m} \mathcal{L}_k^{when} + \mathcal{L}_k^{which} \tag{2}$$

## 3.2 INFERENCE

At inference, we apply a simple softmax function across the decoding step $k$, which produces the zero-shot localizations of the relative temporal intervals of each "Cap", directly from the "When"

Table 1: Qualitative localization & caption generation results of a random sample from YouCook2 validation split, titled $R$ = 'Restaurant Style- Channa Masala' with vid = LeCwqp8Bic8, compared with manual annotations provided by the data set. The the last row statistics are the average of the preceding $m$ rows. A zero-shot captioner DeCap Li et al. (2023b) based on static CLIP embedding is taken as baseline.

| $k = 1, ..., m$ | IOU | "When" $(w_k)$ | | "Which" $(c_k)$ | Bert-score | Clip-score |
|---|---|---|---|---|---|---|
| 1 Baseline Generation | | \|←Manual boundary→\| | | 'a bowl of carrots and other vegetables' | 0.3767 | 0.4028 |
| Our Generation | 0% | \|←0:00 | 0:20→\| | 'add the cooked beans to the pan' | 0.4140 | 0.5075 |
| Manual Annotation | | \|←0:53 | 1:32→\| | 'add onions tomatoes and water to a pressure cooker and cook' | | |
| 2 Baseline Generation | | \|←Manual boundary→\| | | 'a bowl of tomatoes, peppers, and onions' | 0.9474 | 0.8693 |
| Our Generation | 0% | \|←0:20 | 0:48→\| | 'blend the onion and tomato in a blender until smooth' | 0.9417 | 0.7681 |
| Manual Annotation | | \|←1:48 | 1:53→\| | 'add the onion and tomato to a blender and blend' | | |
| 3 Baseline Generation | | \|←Manual boundary→\| | | 'a pot with red powder sitting on a table' | 0.2974 | 0.8982 |
| Our Generation | 0% | \|←0:48 | 1:09→\| | 'add the ghee and oil to the handi' | 0.6232 | 0.7873 |
| Manual Annotation | | \|←2:10 | 2:55→\| | 'add oil, cumin seeds, ginger garlic paste, garam masala, coriander powder, chili powder, and turmeric powder to a pan' | | |
| 4 Baseline Generation | | \|←Manual boundary→\| | | 'a bowl of vegetables in a kitchen ' | 0.2273 | 0.2664 |
| Our Generation | 0% | \|←1:09 | 1:36→\| | 'stir in the tomato and onion blend into the pan' | 0.8325 | 0.6953 |
| Manual Annotation | | \|←2:56 | 3:03→\| | 'add the tomato and onion paste to the pan' | | |
| 5 Baseline Generation | | \|←Manual boundary→\| | | 'add the chickpeas to the pan' | 0.9206 | 0.7258 |
| Our Generation | 0% | \|←1:36 | 2:15→\| | 'pour in the cooked chickpeas to the pan' | 0.8510 | 0.6956 |
| Manual Annotation | | \|←3:18 | 3:28→\| | 'add the cooked chickpeas to the pan' | | |
| 6 Baseline Generation | | \|←Manual boundary→\| | | 'food being cooked in a saucepan' | 0.0988 | 0.7102 |
| Our Generation | 0% | \|←2:15 | 2:53→\| | 'soup is cooked in a pan' | 0.2985 | 0.7265 |
| Manual Annotation | | \|←3:40 | 3:49→\| | 'add dried fenugreek leaves to the pan' | | |
| 7 Baseline Generation | | \|←Manual boundary→\| | | 'a close up of a person stirring a pot of soup' | 0.0000 | 0.1391 |
| Our Generation | 9.52% | \|←2:53 | 4:38→\| | 'mix in the sweetened condensed milk with the other ingredients in the pan.' | 0.8590 | 0.7620 |
| Manual Annotation | | \|←4:20 | 4:30→\| | 'add sweetened condensed milk to the pan' | | |
| 8 Baseline Generation | | \|←Manual boundary→\| | | 'a pot of curry with vegetables and cilantro' | 0.8747 | 0.7729 |
| Our Generation | 18.75% | \|←4:38 | 5:42→\| | 'food cooking in a pot on a stove' | 0.3655 | 0.4136 |
| Manual Annotation | | \|←4:50 | 5:02→\| | 'garnish with coriander leaves and lime juice' | | |
| (Avg) Baseline Generation | | \|←Manual boundary→\| | | | 0.4679 | 0.5981 |
| (Avg) Our Generation | 3.53% | \|←0:00 | 5:42→\| | | 0.6482 | 0.6695 |

- Words in 'green' indicates frequent named entities that occur in both the asynchronous cross-context speech transcriptions of the corresponding sample, and the synchronous cross-modal manual captions;

- Words in 'red' indicates semantic entities / activities generated by the captioners of our / baseline methods but have not occurred in the speech transcriptions of the corresponding sample;

- Words in 'blue' indicates frequent named entities that occur in both the asynchronous cross-context speech transcriptions of the corresponding sample, and the synchronous cross-modal zero-shot output captions, but have not occurred in the manual captions provided by the data set.

embedding $\|W_k\|$, without any beam search needed:

$$w_k = w_{k-1} + softmax(\|W_k\|) \cdot T$$
$$= w_{k-1} + \frac{\exp(\|W_k\|)}{\sum_{k'}^{m} \exp(\|W_{k'}\|)} \cdot T$$

where $w_m = T$ is naturally satisfied under a $softmax$ distribution.

For caption generation at each location $k$, we follow the idea of zero-shot image captioners to project the "Which" embedding $V_R^{which}$ back into the CLIP embedding space, and train a zero-shot text decoder, in the same way as DeCap Li et al. (2023b), to generate sentences conditioned on $V_R^{which}$, which performs exactly a reverse process of the CLIP encoder.

Note that the entire setup in this work, from pre-training to prediction, is fully automatic, does not involve any forms of manual supervision, nor fine-tuning on any downstream objectives.

## 4 EXPERIMENTS

We use the YouCook2 Zhou et al. (2018) and CrossTask Zhukov et al. (2019a), and follow the standard splits for training and testing. The same prompting strategy from the original CLIP publication

Table 2: The semantic generation results $C_R$ on YouCook2 / CrossTask date sets. Best results in each group are highlighted in bold.

| Method | | | | Encoder | Decoder | YouCook2 | | CrossTask | |
|---|---|---|---|---|---|---|---|---|---|
| | | | | | | Bert-Score | Clip-Score | Bert-Score | Clip-Score |
| Zero-shot, with manual localizations $W_k = W_k^{manual}$ | | | | | | | | | |
| 2021 Clip prefix for image captioning Mokady et al. (2021) | | | | CLIP Vit-B/32 | GPT-2 (fine-tuned) | 24.95 | 33.78 | **29.78** | 35.60 |
| 2022 Zero-shot image-to-text generation for visual-semantic arithmetic Tewel et al. (2022) | | | | CLIP ViT-B/32 | GPT-2 (off-the-shelf) | 24.33 | 38.17 | 25.41 | 38.71 |
| 2023 Decoding clip latents for zero-shot captioning via text-only training Li et al. (2023b) | | | | CLIP Vit-B/32 | GPT-2 (trained from scratch) | 25.22 | 37.47 | 28.52 | 33.95 |
| Ours | Initialization | $\mathcal{L}_k^{when}$ | + $\mathcal{L}_k^{which}$ | CLIP Vit-B/32 | $Dec^{which}$ + DeCap (trained from scratch) | 25.22 | 37.47 | 28.52 | 33.95 |
| Ablation (a) = DeCap | Manual | ✗ | ✗ | | | **27.94** | **41.51** | 29.54 | **42.08** |
| Ablation (b) | Manual | ✓ | ✓ | | | $\Delta_a = +2.72 \uparrow$ | $\Delta_a = +4.04 \uparrow$ | $\Delta_a = +1.02 \uparrow$ | $\Delta_a = +8.13 \uparrow$ |
| Fully Zero-shot, joint localization + caption generation | | | | | | | | | |
| Ablation (c) | ✓ | ✗ | ✗ | CLIP Vit-B/32 | $Dec^{when}$ + $Dec^{which}$ + DeCap (trained from scratch) | 12.22 | 21.19 | 14.96 | 28.25 |
| Ablation (d) | ✓ | ✓ | ✗ | | | 16.08 $\Delta_c = +3.86 \uparrow$ | 23.89 $\Delta_c = +2.70 \uparrow$ | 18.73 $\Delta_c = +3.77 \uparrow$ | 29.52 $\Delta_c = +1.27 \uparrow$ |
| Ablation (e) | ✓ | ✗ | ✓ | | | 17.34 $\Delta_c = +5.12 \uparrow$ | 25.76 $\Delta_c = +4.57 \uparrow$ | 19.51 $\Delta_c = +4.55 \uparrow$ | 29.46 $\Delta_c = +1.21 \uparrow$ |
| Ablation (f) | ✓ | ✓ | ✓ | | | 17.92 $\Delta_b = -20.02 \downarrow$ | 26.62 $\Delta_b = -24.89 \downarrow$ | 19.58 $\Delta_b = -15.96 \downarrow$ | 20.55 $\Delta_b = -21.53 \downarrow$ |

- As current best zero-shot caption methods Mokady et al. (2021); Tewel et al. (2022); Li et al. (2023b) are not capable of localization nor report results on the cooking-specific data sets in their original publications, their results are collected by our implementation using the official codebase whose scores are averaged across $m$ frames selected at manually annotated locations provided by the data set.

Table 3: The temporal semantic localization results (mIOU) on the YouCook2 / CrossTask date sets.

| Method | | | | Encoder | Decoder | YouCook2 | CrossTask |
|---|---|---|---|---|---|---|---|
| Fine-tuned | | | | | | | |
| 2021 TSP: Temporally-Sensitive Pre-training of Video Encoders for Localization Tasks Alwassel et al. (2021) | | | | R(2+1)D-34 | Linear Classification | 21.4 | 44.0 |
| Zero-shot | | | | | | | |
| Ours | Initialization | $\mathcal{L}_k^{when}$ | + $\mathcal{L}_k^{which}$ | CLIP Vit-B/32 | $Dec^{when}$ | | |
| Ablation (c) | ✓ | ✗ | ✗ | | | 6.7 | 29.6 |
| Ablation (d) | ✓ | ✓ | ✗ | | | 12.2 $\Delta_c = +5.5 \uparrow$ | 30.5 $\Delta_c = +0.9 \uparrow$ |
| Ablation (e) | ✓ | ✗ | ✓ | | | 10.3 $\Delta_c = +3.6 \uparrow$ | 31.3 $\Delta_c = +1.7 \uparrow$ |
| Ablation (f) | ✓ | ✓ | ✓ | | | 17.0 $\Delta_c = +10.3 \uparrow$ | 35.8 $\Delta_c = +6.2 \uparrow$ |

1. As modern zero-shot image-to-text caption methods are not capable of semantic localization, we choose the current best fully supervised method Alwassel et al. (2021) as the baseline.
2. The baseline result is collected by our implementations using the official codebase, bacause the Alwassel et al. (2021) uses a retrieval based metric mAP which is not originally designed for semantic localization, not as intuitive nor explainable as mIOU.

Radford et al. (2021) is applied on CrossTask data set for sentence level embedding since there are no sentence level annotations provided in CrossTask data set for Bert score calculation. All encoders / decoders have the same 12 layers, 8 heads, embedding dimension 512, and MLP hidden dimension of 2048. There are 314M trainable parameters in total. For training we use AdamW with fixed weight decay, with a learning rate of $10^{-4}$ and 1000 warm-up steps. We pretrain our model for 8 epochs with a batch size of 32 videos split on 8 V100 GPUs, which lasts 1.6 days.

Table 2 and 3 reports captioning & localization performance of our method as compared to the state of the art, respectively. Under the same setup of manual locations $W_R = W_R^{man}$, our method achieves absolute [Bert, Clip] scores of [27.94, 41.51] / [29.54, 42.08] on YouCook2 / CrossTask, respectively, consistently outperforming the state of the art baselines with $\triangle_{baselines} = [+2.72 \uparrow, +3.34 \uparrow]$ / $[-0.24 \downarrow, +3.37 \uparrow]$ points higher. For semantic localization, our method achieves competitive overall mIOU in a fully zero-shot way, only $\triangle_{baseline=TSP} = 4.4\%/8.2\%$ absolute gap behind the baseline Alwassel et al. (2021) on YouCook2 / CrossTask, respectively.

From a higher-level perspective, Table 2 and 3 jointly verifies our initial motivation: 1. from the upstream perspective, the integration of $Dec^{when}$ promotes the generation accuracy of "Which", and vice versa; 2. from the downstream perspective, the higher accuracy in localization $W_R$ necessarily associates with a higher accuracy in caption generations $C_R$, and vice versa; which inspires us the necessity of "Which" + "When" joint embedding: the semantics of "Which" and the contexts of

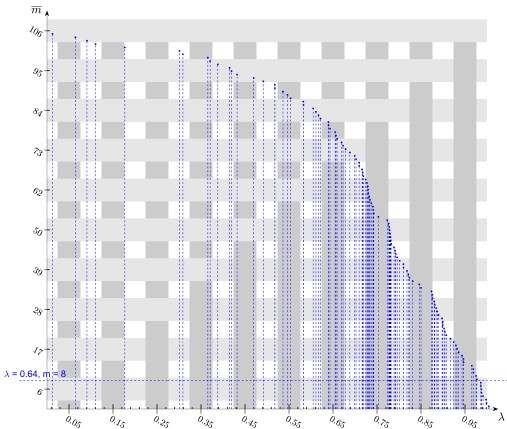

Figure 4: How the value of $\overline{m}$ changes over hyper-parameter $\lambda$ on YouCook2 Zhou et al. (2018).

"When" are inseparable for each other, mutually promoting each other and missing a piece without each other for comprehensive semantic understanding.

## 4.1 $\lambda$ SELECTION DETAILS

In order to facilitate customizable semantic density $\overline{m}$ on different data sets, we introduce a hyper-parameter $\lambda$ for different data sets: Figure 4. See Eq. equation 1 in section 3.1 for the relationships between the average number of outputs captions $\overline{m}$ and the semantic similarity threshold $\lambda$. For a specific downstream task like cooking video caption, $m$ on average is usually an order of magnitude less than the number of speech sentences $n$. For instance the YouCook2 Zhou et al. (2018) data set contains 1,790 videos with 132, 310 (clip $\leftrightarrow$ sentence) pairs, with average statistics being:

- $\overline{n} = 73.9/$ per video;
- $\overline{m} = 7.7/$ per video;
- $\overline{T} = 320$ seconds / pre video;

## 4.2 VISUALIZATIONS

Figure 3 and Table 1 show a random example from the YouCook2 data set. Quite a number of interesting findings can be observed:

### 4.2.1 TEMPORAL VISUAL EMBEDDING (OURS) $\neq$ STATIC VISUAL EMBEDDING (BASELINE):

compared with the baseline, our method achieves higher overlaps on non-stop words with manual annotations, on average, especially on named entities (nouns) / activities (verbs) that frequently appears in asynchronous / synchronous speech utterances from the temporal context (for example, "oil", "tomato", "onion", "blend", and "condensed milk", highlighted in "green" at $k = 3, 4, 7$ in Table 1). This indicates the ability of our temporal visual embedding on cross-context and cross-modal joint reasoning, which cannot be achieved from a static embedding.

### 4.2.2 MANUAL ANNOTATION $\neq$ GROUND TRUTH:

We also notice that a low Bert score does not necessarily means bad performance, because manual annotations do not always comply with the ground truth in a challenging dense video caption task. For example, we get a low Bert score = 0.4140 at $k = 1$ in Table 1, however it is caused by the fact that our method successfully discovers a non-negligible step omitted in the manual annotation provided by the data set, which is $c_1 = $ 'add the cooked beans to the pan'. $c_1$ is both visually and linguistically salient and indispensable for the overall task of making the 'restaurant style channa masala'.

### 4.2.3 BERT SCORE ≠ CLIP SCORE:

Bert score Zhang* et al. (2020) is more prone to location inconsistence as compared to Clip score Hessel et al. (2021), a satisfactory Clip score with a deteriorated Bert score occasionally happens for example, $k = 1, 6, 8$ in Table 1. This observation is consistent with our quantitative results, that our approach favours the evaluation metric of Clip score more than Bert score, former of which, is more objective as Clip score is fully reference-free, does not involve any forms of subjective bias introduced by manual annotations from the data set.

### 4.2.4 MORE CHALLENGING ≠ LESS ACCURATE:

From a common instinct, the task of dense video caption should be significantly more difficult than that of image caption, and its zero-shot performance, should be even worse given the complication introduced by the temporal dimension. However, the fact goes the opposite way against intuitive expectations in both the quantitative and qualitative results: the performance gap between our zero-shot dense video captioning and fully supervised baselines is substantially less obvious than that between zero-shot image captioners and their fully supervised counterparts. This is another inspiration which justifies our initial motivation, that rather than bringing complexities, what a joint embedding really brings are more resourceful information, more complementary self-supervision, less uncertainties, higher confidences, and last but not the least, a more robust way toward comprehensive semantic understanding.

## 5 CONCLUSION

In this work, we facilitate dense cooking video recipe generation by learning a contextualized CLIP embedding, from natural language supervision. Our method is simple, data-scalable, open-vocabulary, zero-shot and at a low computation cost, however there do come along with some limitations worth mentioning: 1. the first disadvantage lies in that we replace the common video processing unit of clips with frames, resulting in a lack of spatiotemporal representation, however this is not a crucial disadvantage because short-term motion plays a very minor role in long-term semantic understanding. 2. the second disadvantage lies in the reliance on the ASR transcriptions, videos without speech transcriptions, or with very weak vision ↔ speech semantic associations do not provide effective supervision, however this is also not a crucial disadvantage in cooking domain because training data with satisfactory cross-modal speech association is plentiful, and once upon the convergence of the training stage, our embedding is zero-shot transferrable to vision-only modality and fully free from the constraint at the inference stage. Further explorations might also include better interpretability of the learned embedding, from a more theoretical perspective, which we leave for future efforts.

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

## A APPENDIX

### A.1 DETAILS ON THE DATA PREPARATION FOR TRAINING

#### A.1.1 THE TITLE $R$ OF THE VIDEO:

In the from of natural language, $R$ is composed as either several words / phrases / short sentences, which can also be fed to the off-the-shelf pre-trained language encoders, like Bert / CLIP, for [CLS]

tokenization in our embedding. For example $R =$ "`Restaurant Style Channa Masala`" for $V_R$ at `www.youtube.com/watch?v=LeCwqp8Bic8`.

### A.1.2 THE FRAME SEQUENCE $V_R$:

The video $V_R = \{v_1, ..., v_n\}$, lasting $T$ seconds in length, and segmented into $n$ continuous clips accordingly to align with the sentences in the transcript $S_R$, with the title $R$. For computational simplicity, one and only one frame is selected as the visual representation for each clip, which is formally equivalent to $v_i$ in the context of this work.

### A.1.3 THE SENTENCE SEQUENCE $S_R$:

The temporal sequence of $n$ sentences $S_R = \{s_1, ..., s_n\}$ representing $V_R$'s corresponding transcripts. Not every clip $v_i$ aligns to a virtual utterance, as common videos include non-speech segments like an intro, an outro, in-between transitions, and background musics. We introduce a background $s_i = SIL$ to align with silent clips $v_i^{SIL}$ without speech transcriptions.

### A.1.4 THE START / END TIME STAMPS SEQUENCE $T_R$ FOR BOTH $V_R$ AND $S_R$:

To guarantee the temporal alignment between frame $v_i \leftrightarrow$ sentence $s_i$ pairs, $V_R$ are segmented with the Google Cloud API [1], with its corresponding time stamps $T_R = \{t_1^{start}/t_1^{end}, ..., t_n^{start}/t_n^{end}\}$ of each transcribed speech sentence $s_i$. Note that with the introduction of the silent (clip $\leftrightarrow$ sentence) pairs $\left(v_i^{SIL} \leftrightarrow s_i = SIL\right)$, we facilitate seamless processing both at the front ($t_i$) and back ($w_k$) end:

$$
t_i = \begin{cases}
t_i^{start} = 0 & i = 1 \\
t_i^{end} = t_{i+1}^{start} & i = 1, ..., n-1 \\
t_i^{end} = T & i = n
\end{cases}
$$

$$
w_k = \begin{cases}
w_k^{start} = 0 & k = 1 \\
w_k^{end} = w_{k+1}^{start} & k = 1, ..., m-1 \\
w_k^{end} = T & k = m
\end{cases}
$$

## A.2 DATA SETS DETAILS

### A.2.1 YOUCOOK2

YouCook2 Zhou et al. (2018) consists of 1,790 videos of 89 cooking procedures (eg., spaghetti and meatballs) from YouTube. The videos were separated into a 67% / 23% / 10% percent for training / validation / testing split and categorized by humans into one of 89 recipe types. Videos were temporally segmented by human annotators into clips $v_i$ representing recipe steps, and each clip was annotated with a text summary $s_i$ of the recipe step. On average, each video lasts $\overline{T} = 320$ seconds and is annotated with $\overline{m} = 7.7$ temporally-localized imperative sentences. Following Miech et al. (2019), we use 9,586 training clips and 3,350 validation clips due to the unavailability of some videos on YouTube.

### A.2.2 CROSSTASK

CrossTask Zhukov et al. (2019b) consists of 2,750 instructional videos from YouTube with 18 primary tasks and 65 related tasks. On average, each video lasts $\overline{T} = 297$ seconds and is annotated with $\overline{m} = 7$ temporally-localized steps, such as "`remove cap`" and "`spread mixture`". 20 videos from each of the 18 primary tasks are designated as the validation set (360 videos total), and the rest are left for training. The videos in the validation set were temporally localized into clips $v_i$ for each step by human annotators, while the videos in the training set were localized into clips for each step automatically based on the ASR transcripts. The training set contains 18,067 clips while the validation set contains 2,852 clips.

---

[1] `https://cloud.google.com/speech-to-text/docs/automatic-punctuation.`

### A.3 EVALUATION DETAILS

#### A.3.1 EVALUATION DETAILS FOR SEMANTIC GENERATION

Similar to image to text caption, we apply two measurements for temporal semantic generation quality assessment: one reference-based (Bert score Zhang* et al. (2020)) and one reference-less metric (Clip score Hessel et al. (2021)).

**Reference-based metric** Bert score uses a pre-trained bert Devlin et al. (2019) model to calculate the cosine similarity between the human-created texts embedding $\mathbf{C}^{ref}$ and the texts embedding generated by the model $\mathbf{C}$, associated with the image.

$$Bert - score = sim(\mathbf{C}, \mathbf{C}^{ref}) = \frac{\mathbf{C} \cdot \mathbf{C}^{ref}}{\|\mathbf{C}\| \cdot \|\mathbf{C}^{ref}\|}$$

**Reference-less metric** CLIP score uses a pre-trained CLIP Radford et al. (2021) model to calculate the cosine similarity between the image embedding $\mathbf{V}$ and the caption embedding $\mathbf{C}$ without the need for any references.

$$Clip - score = sim(\mathbf{V}, \mathbf{C}) = \frac{\mathbf{V} \cdot \mathbf{C}}{\|\mathbf{V}\| \cdot \|\mathbf{C}\|}$$

#### A.3.2 EVALUATION DETAILS FOR SEMANTIC LOCALIZATION

we apply $mIOU = \frac{1}{m} \sum_k^m IOU_k$ for temporal semantic localization quality assessment, which is calculated by the following three steps of at progressive granularity:

1. The $IOU_k$ for each ground-truth segment $k$ within an individual video $V_R$: map the predicted indices $w_k$ into the corresponding ASR token time stamps and compare it against the segment's manual start and end time stamps. Recall that the ground-truth segments are marked by start (and end) times, whereas the predicted segments are expressed according to the position of the corresponding ASR token, and an IoU score can be computed for each (ground-truth, predicted) segment pair.

2. The video-level $mIOU$ for each video: $mIoU$ is the average of $IOU_k$ across all ground-truth segments. The video-level $mIoU$ provides a summary score for segmentation performance over the entire video.

3. The overall $mIOU$: the individual $mIoU$ for each video is then averaged across the test data split and reported as the overall $mIoU$.

