# OpenReview forum: "CookingCLIP：Learning a Contextualized Multimodal Embedding from Instructional Cooking Videos for Zero-shot Recipe Generation"
_ICLR.cc/2024/Conference — ICLR 2024 Conference Withdrawn Submission_

### Official Review · Reviewer_f8ZN · 2023-10-24

**Soundness:** 3 good
**Presentation:** 2 fair
**Contribution:** 3 good
**Rating:** 5
**Confidence:** 3

**Summary:**

The paper generates recipes with timestamps for each step from a video. Their architecture composes a pair of original CLIP encoders for each modality, each of which being extended with a pair of auto-regressive decoders for localization and caption generation. Experiments on YouCook and CrossTask demonstrate the effectiveness of the proposed embedding.

**Strengths:**

1. Their paper both generates the natural language steps and  temporal localizations for each step.
2. Their method outperforms the baselines in terms of Bert-Score and CLIP-Score.

**Weaknesses:**

1. There notations are not well-defined in the paper. Different variables use the same annotation, e.g. W can be timestamps (Experiment section),  embeddings (eq (1)) and tokens (Wm=EOS), which is very confusing. wk is not defined or used later either, only the recurrence function is given.
2. It seems temporal localization is a significant bottleneck of the method. Both from the qualitative examples and Table 3, where the mIoU of the proposed model is significantly lower than the baseline. Action detection/localization models could also be included as baselines. The annotated step timesteps are included in the dataset, ASR can include dense timestamps and there are datasets with dense timestamp annotation for each step/sentence (EPIC-KITCHENS), which can be utilized.
3. Bert-Score and CLIP-Score needs to be justified. E.g. In table 1 step 6, the difference of CLIP score between the methods is much smaller than Bert-Score. Traditional scores like BLEU-4, METEOR, CIDEr, and ROUGE also needs to be reported. Metrics focusing on actions and entities can also be reported.

**Questions:**

1. Figure 4, why m seems to decrease as \lamda increases? Which is counterintuitive according to eq (1)
2. In Section A1.1.1, where is the title used in the method?

---

### Official Review · Reviewer_Qj9E · 2023-10-27

**Soundness:** 1 poor
**Presentation:** 1 poor
**Contribution:** 2 fair
**Rating:** 3
**Confidence:** 4

**Summary:**

This paper proposes a method to pre-train vision-language models from instructional video. The method aims to extend a clip-like model with temporal context by explicitly dividing time-series features into `when` and `which.`
The pre-trained model is evaluated on YouCook2 and CrossTask datasets with a dense video captioning task, but only with the author’s proposed metrics.

**Strengths:**

- The motivation is clear.
- The result looks slightly better than the baselines in the manual initialization setting.

**Weaknesses:**

1. The evaluation protocol is not well justified.

The authors proposed a new metric, but no traditional metrics for text generation, but only those for latent feature extractions. Since these tasks are traditionally evaluated on metrics for text generation, the authors are responsible for justifying the new metrics. A deeper experimental analysis is necessary to switch from a traditional evaluation protocol.

2. Unclear presentation

First of all, the authors never define mathematical notations in the text, but only in Figure 2. Since the definition heavily relies on the user’s understanding of visual information, they are not rigorous and hard to understand correctly. For example, this reviewer does not understand the difference between S_m^*. S’_m, and S_m^{which}. This unclarity was critical for this reviewer because it is impossible to guess the difference between $L^{when}$ and $L^{which}$; these are the core ideas of this method. The other figures also have too many texts (Fig. 1, 3, Table 1), which is not an ordinal paper format. Fig. 3 has too many unreadably tiny fonts, and sometimes they overlap. These texts may violate the ICLR format.

**Questions:**

Please point out any factual errors in this review if the authors find them.

---

### Official Review · Reviewer_1XSR · 2023-11-01

**Soundness:** 2 fair
**Presentation:** 1 poor
**Contribution:** 1 poor
**Rating:** 3
**Confidence:** 4

**Summary:**

This paper addresses the strong cross-modal and cross-context dependencies characteristic of cooking videos by making two significant adaptations to the original CLIP embedding: firstly, by extending static multi-modal CLIP embedding with a temporal dimension for context-aware semantic understanding, and secondly, by introducing zero-shot embedding to sequence-to-sequence dense prediction domains. This innovative approach enables CookingCLIP to not only recognize "Which" (cross-modal recognition) but also determine "When" (cross-context localization). The efficacy of CookingCLIP is demonstrated through experiments on challenging cooking caption generation benchmarks, YouCook and CrossTask, showcasing its effectiveness in the domain.

**Strengths:**

1. The proposed idea is straightforward and intuitive.
2. The targeted problem is interesting.

**Weaknesses:**

1. The contribution is limited.

a. The technical innovation is to add a location prediction branch in the decoding process based in automatically generated weak supervision. This idea itself of joint localization and understanding is not really new to this domain. The content decoding branch is also built upon existing work DeCap.

b. The provided insight is also somewhat limited. The authors leave the truly interesting part: why does the learning process of location helpful for improving the caption generation as future research. However, the true insight behind these behavior is still missing. It is not clear whether the location is really the key here or the key is actually to **perform domain adaptation of the CLIP embedding on the cooking videos**. To really understand this, the authors should have provided more thorough comparison with other baselines like simply finetuning/or training adapters based on CLIP on the cooking videos without formulating the when loss but use other loss formulation like masked token modeling, regular contrastive learning and etc.

c. The contribution is also over-claimed. It is not appropriate to claim that this work is "zero-shot, open-vocabulary, and fully free from manual supervision". The current limited domain itself has already set the limitation of the generalization. Moreover, the zero-shot in this paper is exactly defined based on a close vocabulary of cooking recipes, which is defined in each of the datasets.

2. The presentation really needs further improvement.

a. Definition of terminology is confusing. In the reviewer's view, this is nothing about recipe generation. A recipe should be a general sequence of steps to making a meal but this model is really a video content translator and localizer.

b. Almost all the figures and the tables contain way too much details. Some of them are difficult to read even when zoomed in on a 27'' display. The authors should really reduce details and make the key message clearer on all of the figures and tables by reducing details.

3. Further baselines like SOTA models VidSeq for caption generation with Google Cloud automatic segmentation or VideoCLIP as the localizer should be provided to help the audience understand better about the performance.

**Questions:**

Please check weakness for details.

---

### Official Review · Reviewer_5T4L · 2023-11-01

**Soundness:** 2 fair
**Presentation:** 1 poor
**Contribution:** 1 poor
**Rating:** 1
**Confidence:** 3

**Summary:**

The paper introduces CookingCLIP, which adapts the CLIP (Contrastive Language-Image Pre-training) embedding concept from the general domain to the specific domain of cooking understanding. The authors enhance the original image-based vision-language contrastive learning by incorporating a temporal dimension, presenting this as the technical novelty. Furthermore, they propose the zero-shot recipe generation task, evolving from the traditional dense video captioning benchmark.

**Strengths:**

The paper endeavors to introduce a CLIP-like multimodal embedding tailored for zero-shot cooking understanding, presenting an overarching concept that is fundamentally sound at a high level.

**Weaknesses:**

The paper appears to be poorly articulated, leading me to suspect that it may be an incomplete submission.

**Questions:**

1. What is $T_R$ in the beginning of Sec. 3? How about $W^V_k$, $W^S_k$, $V^{which}_k$, and $S^{which}_k$? These notions are not defined.

2. Which part of Sec. 3 details the technical novelty in temporal modeling?

3. What is “semantic density” described in Sec. 4.1?

4. Could you explain the process for constructing positive and negative pairs?